# Fear/Anxiety and Sleep Deprivation Combine to Predict Courage

**DOI:** 10.3390/bs15050634

**Published:** 2025-05-06

**Authors:** Jeffrey A. Gibbons, Brenna E. McManus, Ella C. White, Akihaya M. Gibbons

**Affiliations:** 1Department of Psychology, Christopher Newport University, 1 Avenue of the Arts, Newport News, VA 23606, USA; brenna.mcmanus.22@cnu.edu (B.E.M.); ella.white.22@cnu.edu (E.C.W.); 2Department of Psychology, Virginia Polytechnic Institute and State University, Blacksburg, VA 24061, USA; akig@vt.edu

**Keywords:** courage, anxiety, fear, sleep quality and quantity

## Abstract

The current study examined the combined effects of sleep deprivation and anxiety on participants’ willingness to act courageously in both heroic and everyday situations. The participants consisted of 256 undergraduate students between the ages of 18 to 25 years old seeking regular and extra credit for their psychology classes through SONA. Following informed consent, the participants completed demographic questionnaires through Qualtrics, as well as the Depression Anxiety Stress Scale, the Pittsburgh Sleep Quality Index, and an adapted version of the Woodard Pury Courage Scale-23 (WPCS-23). The adapted Woodard Pury Courage Scale-23 measures participants’ willingness to engage in challenging tasks that require either heroic or everyday courage and the fear they would experience when engaging in these tasks. The six measures of courage included willingness to engage in everyday, heroic, and both acts, as well as fear when engaging in these actions. Fear/anxiety by sleep interactions predicted every courage measure except for fear when engaging in daily courageous actions. The results supported the hypothesis that fear/anxiety and poor sleep would combine to predict courage, and their implications are discussed.

## 1. Fear/Anxiety and Sleep Deprivation Combine to Predict Courage

Few people choose to jump out of a perfectly functioning airplane or off a cliff, and even fewer individuals run into a burning building or throw themselves into the fray to help strangers in precarious positions daily, because these potentially harmful situations are scary, and they challenge the self-preservation instinct. Although heroism is rare, courage is displayed every day when people ask strangers for directions, stand up for themselves or someone else, or ask a question in a crowd. Courage is the act of overcoming the hardwired fear experienced by all animals that guides them to avoid or escape dangerous situations that can cause them harm ([32]; [33]). In fact, acts likely cannot be truly courageous in the absence of fear, even though high levels of fear clearly reduce courage ([33]; [34]). In addition, sleep-deprived individuals may be more likely to engage in risky situations than sleep-sufficient individuals, because sleep deprivation hinders mental clarity and decision-making ([38]). For example, [23] ([23]) found that military personnel engaged in courageous actions when they were heavily sleep deprived. Although research has demonstrated that fear is related to courage and sleep deprivation enhances courage-enabling risky decision-making, no study has evaluated the combined relation of fear and sleep deprivation as an interactive predictor of courage. Therefore, the current study examined the relations of fear, as measured by anxiety, poor sleep (both quantity and quality), and their combination as predictors of everyday and heroic acts of courage.

### 1.1. Fear/Anxiety and Courage

[14] ([14]) determined that fear, in addition to purpose, agency, and availability, was viewed as a precondition for courage. [31] ([31]) defined courage as the willingness to act in the presence of subjective fear, and [7] ([7]) stated that fear is the driving factor of courage, making it the act of overcoming fear. Courage has also been defined as the ability to overcome one’s fear when encountering obstacles that involve stressful or dangerous conditions ([33]), and [33] ([33]) stated that traditional courage involves overcoming fear, as well as facing obstacles and risk. [8] ([8]) found that seven decorated bomb-disposal operators performed more accurately and calmly in a stressful test than seven undecorated bomb-disposal operators and seven civilian subjects in the comparison groups. Similarly, [25] ([25]) found that 105 paratrooper recruits in training reported increased confidence and decreased fearfulness over multiple jumps. More recently, [4] ([4]) suggested that courage buffers individuals from feeling overly scared in very tense situations, helping them manage stress and anxiety. [34] ([34]) highlighted the importance of fear and anxiety for determining the presence of courage; they asserted that fear and anxiety foster courage by challenging individuals to act despite discomfort.

[27] ([27]) defined courage as action in the presence of fear, which emphasizes the importance of fear for the existence of courage. These researchers then examined the relationship between courage, anxiety, and fear using a behavioral approach. The sample consisted of 31 female undergraduate psychology students (mean age of 22.13 years) who participated in a two-part study. The first part of the study measured spider fears via the Spider Questionnaire ([21]) and the Spider Phobia Beliefs Questionnaire ([3]). The second part of the study evaluated courage, as measured via a researcher-developed courage scale ([34]), anxiety, as measured via the State-Trait Anxiety Inventory ([36]), and distress, as measured using the Subjective Units of Distress Scale ([40]). The participants were then presented with a spider, and their spider fears were evaluated using a Behavioral Approach Test (BAT), which asked the participants to place their hand as close to the spider as they were comfortable doing. Distance from the spiders was the fear/courage measure evaluated in the behavioral approach, with far distances indicating high fear and low courage. The results showed significantly lower fear and higher courage during part one than in part two of this study, which suggested that the willingness to engage in courageous acts is strongly and negatively influenced by the degree of fear elicited by the situation.

[6] ([6]) replicated the spider study, but they focused on the BAT as the main measure of fear/courage and evaluated the effect of expressed task importance on it. Again, long distances indicated high fear and low courage, and the participants were randomly assigned to task performance conditions using monetary incentives. The low-task-importance group was simply told that they would receive a USD 20 participation reward. The high-task-importance group was told that they would receive a maximum amount of USD 20 for their participation, but the amount received depended directly on how close they became to the spider, with each additional 3 cm moved towards the spider earning an additional USD 1. Following the BAT, the participants completed the Courage Measure (CM; [27]) and the Subjective Units of Distress Scale (SUDS; [40]). The researchers found that large taxidermy tarantulas induced fear as every participant did not touch the fake spider, and, more importantly, distances from the spider were shorter for the high-task-importance group than the low-task-importance group. This result showed that fear plays an important weakening role in displays of courage. However, CM scores predicted BAT distance above and beyond distress, which indicates that courage helps individuals act in ways that help them overcome their fears.

Using the same fear-provoking spider task as the previously described studies, [12] ([12]) examined the effect of a positive psychology intervention, known as exposure therapy, on acts of courage. The sample consisted of 96 undergraduate students (69.4% female) who were tested on their fear of spiders. Fear of spiders was measured using an adapted version of the Fear of Spiders Questionnaire ([37]), and courage was measured using the Courage Measure (CM; [27]). The participants were randomly assigned to a control condition or a positive psychology intervention, which involved an experimenter directly asking them several therapeutic-style questions regarding a scenario in which they encountered spiders, pertaining specifically to the behavior. The control group heard another series of questions regarding a scenario in which they encountered spiders and focused on their current feelings.

For all the participants in the [12] ([12]) study, the therapeutic intervention was followed by a sorting task. All participants then completed an exposure analogue in which they were asked to touch the live spider in different ways. After engaging in each method of touching the spider, in which closeness to the spider was measured, the participants reported courage scores using the CM. Each interview was rated for fearfulness (from 1 to 10), approach behavior (from 1 to 10), and courage (fear × approach/100). The results showed that the treatment intervention was effective in producing greater approach behavior and courage when the interview fear ratings were used as a covariate. Much like the results of [6] ([6]), these results suggested that true courage helps individuals act in a way that allows them to overcome their fears.

Based on the literature showing that fear is a critical component of courage, the current study used an adapted version of the Woodard Pury Courage Scale 23 (WPCS-23; [41]) because it used agreement and fear ratings to evaluate courage and fear when engaging in courageous acts, respectively. Specifically, the participants were provided with actions that would demand courage to engage in, and they were asked to provide the degree they would agree to engage in the action as well as the fear they would experience when engaging in the action. Although the original version of the WPCS contained courageous actions described by [29] ([29]) as general courage (courageous for everyone) and personal courage (courageous for the individual), the adapted version of the WPCS examined heroic and everyday courageous acts. Heroic actions were evaluated by statements, such as “I would undergo physical pain and torture rather than tell political secrets”, whereas everyday actions were evaluated by statements, such as “I would seek out and ask a grocery store employee for help when shopping”.

### 1.2. The Relations of Risky Decisions to Courage and Sleep Deprivation

The courage literature and the sleep deprivation literature do not examine or describe a relation to the other body of work. However, each topic involves risky decision-making. As previously stated, courage is defined by risk ([22]; [38]). In addition, sleep deprivation leads to poor/risky decisions ([23]). To establish a relation between the two topics, we examined each of their connections to risky/poor decision-making.

#### 1.2.1. Courage and Risky Decisions

Risky decision-making is a critical component of courage ([4]; [27]). [17] ([17]) found, when validating the [27] ([27]) Courage Measure (CM), that the measure of courage was positively and strongly correlated with risk-taking. [16] ([16]) defined courage as ethical risk-taking across a variety of disciplines, including the field of medicine. [30] ([30]) found that ratings of courage increased with risk-taking, but courage ratings were most strongly influenced by the degree to which participants agreed with the actions in scenarios describing LGBT+ issues, including public gender transitioning and same sex marriage.

Risky decision-making is enhanced by large rewards and high stakes. [39] ([39]) found that risk-taking decisions increased with the degree of reward irrespective of whether the scenario was real or contrived. In a literature review, [13] ([13]) argued that many people try to maintain the status quo, but high stakes often necessitate risky decisions irrespective of context (e.g., investment, gambling, or theft). In a meta-analysis, [14] ([14]) determined that large rewards led to high risks and courage. The literature shows that risky decision-making is an important part of courage, and large rewards increase risky decisions and, hence, courage.

#### 1.2.2. Sleep Deprivation and Risky Decisions

Although military personnel are known for their bravery on the battlefield ([4]), they are also known to battle sleep deprivation alongside their enemy, with the outcome being poor decisions ([23]). [23] ([23]) examined the effect of insufficient sleep on military performance in an anonymous survey of 679 Army soldiers deployed to Afghanistan. They discovered that insufficient sleep significantly increased the risk of accidents and mistakes, causing impulsivity, poor judgment, and unjustified risk. Sleep deprivation has been found to result in clouded judgment, decreased critical thinking skills, increased risk-taking, and reduced inhibitions in non-military samples as well ([28]; [38]). For example, sleep deprivation has also been found in university students, who are historically known to be highly sleep deprived and impulsive ([28]). In a study of 803 students evaluating overall anxiety and sleep quality, 74.5% of sampled university students reported poor sleep, which was related to risky behaviors, such as alcohol consumption.

[38] ([38]) conducted a review with 32 different studies evaluating the effects of sleep deprivation on risky decision-making. A positive relationship was found between sleep deprivation and risky decisions in 25 of the 32 studies using various tasks, such as the Ballon Analogue Risk Task (BART). The BART instructed participants to pump up a virtual balloon with the intent of a reward, but each pump increased the risk of the balloon popping, which resulted in the loss of a reward. Sleep-deprived participants were more likely to display risky decision-making in the BART procedure, as well as the Experiential Discounting Task, the Adjusting Amount Delay and Probability Discounting Task, and the Stop Task than non-sleep-deprived participants, with the effect being stronger for men than women. [22] ([22]) used a modified BART to investigate the risk-taking behaviors in sleep-deprived people. Thirty-one adult men with habitually good sleep habits between the ages of 18 and 28 years participated in the study. The analyses showed that sleep-deprived participants pumped more air into the balloon, indicating higher risk-taking, than other participants. Instead of using the BART, [20] ([20]) investigated the relationship between sleep deprivation and risky decisions using the Iowa Gambling Task. The results showed that sleep-deprived men and women selected more cards from high-risk decks on the Iowa Gambling Task than male and female controls; this result was not moderated by sex. [9] ([9]) used the CANTAB battery procedure to examine the effect of sleep deprivation on risky decision-making in 10 healthy men ranging from 24 to 31 years old who completed the CANTAB procedure. The study examined whether sleep deprivation increased reaction time and reduced focus on the CANTAB. The researchers found, using Doppler scans, decreased activity in the left frontal gyrus for sleep-deprived individuals, which indicated lower risk control. Moreover, the participants were more likely to make risky decisions after being sleep-deprived than after being well-rested. The literature examining the relation between sleep deprivation and decision-making consistently showed that sleep-deprived individuals demonstrated more risky decision-making than non-sleep-deprived individuals.

### 1.3. The Current Study

The courage literature suggests that fear/anxiety is part and parcel of courageous actions ([35]) because an act is not very courageous if the thought of engaging in that action does not elicit feelings of escape and avoidance ([33]). As courage is not related to sleep deprivation in the literature, we investigated the links between risky decisions and both courage and sleep deprivation. This research showed that risky decision-making is positively related to both courage ([4]; [27]) and sleep deprivation ([28]; [38]). Therefore, courage and sleep deprivation should be related through risky decision-making, but the courage literature has not established this relation, nor has it examined the combined effect of fear/anxiety and sleep deprivation as an interactive predictor of courage.

The current study was created to fill this void in the literature and evaluate the interactive relationship of fear/anxiety and sleep measures to levels of heroic and everyday courage. As stated previously, we used an adapted version of the WPCS-23 to measure willingness to engage in and fear when experiencing heroic and everyday acts of courage. We expected fear/anxiety to positively predict agreement courage measures and negatively predict fear–courage measures, sleep hours to negatively predict agreement–courage measures and positively predict fear–courage measures, and poor sleep quality to positively predict agreement–courage measures and negatively predict fear–courage measures. Moreover, we expected the product of fear/anxiety and sleep hours and the product of fear/anxiety and poor sleep quality to predict courage measures. We also expected the products of fear/anxiety and sleep to predict courage measures over and above the individual relations of fear/anxiety and sleep deprivation to courage measures.

## 2. Method

### 2.1. Participants

The original sample of the current study consisted of 256 undergraduate students ranging between the ages of 18 and 28 years old from a small liberal arts university in the southeastern United States. A total of 11 participants did not follow directions, or they provided incomplete information, which resulted in a final sample size of 245 participants. All participants were recruited through Sona Systems (SONA) to register student participation credit for regular and extra credit within their psychology classes at their university. Data for this study were collected between fall 2023 and spring 2024. The participants were predominantly women (162 women and 81 men), with 3 participants selecting “other” or “prefer not to say”. Following recruitment, the participants received an external link to complete an online survey through Qualtrics. Following participation, each participant received class credit. This study received approval from the internal review board (IRB) of the university. According to IRB guidelines specified by the American Psychological Association ([2]), the participants were briefed, provided informed consent, and were debriefed by the conclusion of this study. The participants were allowed to leave this study without penalty, and all data were kept confidential to protect their privacy.

### 2.2. Materials and Measures

The materials for the current study included a consent form, demographics questionnaire, the Pittsburgh Sleep Quality Index (PSQI; [5]), the Depression Anxiety Stress Scales (DASS; [24]), a modified version of the Woodard Pury Courage Scale (WPCS-23; [41]), and a debriefing form. The consent form contained a briefing and a general description of procedures, and it provided contact information for the principal investigator, the university’s IRB chair, and university counseling services. The consent form also provided a section for participants to digitally sign their names. The dependent variable in this study was courage, which was measured with the adapted Woodard Pury Courage Scale. The predictors in this study included sleep deprivation, as measured by the PSQI, and fear/anxiety, as measured by the DASS.

#### 2.2.1. Demographic Questionnaire

The participants were asked to select the option that best represented them concerning their race, gender, sexual orientation, religion, and age. All participants were given the option, “Prefer Not to Disclose”, for each section if they wished not to disclose such personal information.

#### 2.2.2. Modified Version of the Woodard Pury Courage Scale (WPCS-23)

The adapted version of the WPCS-23 is a self-report inventory used to measure each participant’s willingness to engage courageously in fear-provoking situations, as well as the resulting fear/anxiety caused by each engagement. The modified list of 20 questions consisted of two parts, with each item providing a scenario (e.g., I would risk rejection by important others for a chance at achieving my life goals). The participants then provided the degree to which they disagreed/agreed with the statement on a scale ranging from 1 (Strongly Disagree) to 5 (Strongly Agree) and indicated the level of fear they would feel in that situation, rated on a scale ranging from 1 (Little Fear) to 5 (Very High Fear). A factor analysis was conducted to determine a two-factor structure with heroic and everyday courage factors.

The 10 heroic courage items (e.g., I would undergo physical pain and torture rather than tell political secrets) included 1–3, 5–9, 11, and 12. The 8 everyday courage items (e.g., I would meet with my supervisor at work about a problem I believed was important) included 4, 10, 13, 14, 15, 17, 18, and 19. The specific changes that the adapted version of the WPCS-23 used in the current study are reflected in Appendix A. Questions 12 and 20 were removed from further analysis as they did not contribute significantly to either courage measures. By adapting WPCS-23, we created items 13, 14, and 16 to 20 specifically for this study (Appendix B); these replaced items 1, 2, 3, 5, 9, 11, 14, 18, 22, and 23 on the original WPCS-23, which were not used in the adapted version of the WPCS-23. These two sets of questions were totaled for agreement courage statements and fear when engaging in these courageous situations, producing four courage measures. The totals were calculated for courage statements and fear statements, producing two additional measures. The Cronbach’s alphas for heroic agreement courage, everyday agreement courage, total agreement courage, as well as fear when engaging in heroic, daily, and both heroic and daily courageous actions were 0.729, 0.704, 0.784, 0.845, 0.828, and 0.895, respectively.

#### 2.2.3. Brief Depression Anxiety Stress Scale (DASS-21)

The brief Depression, Anxiety, and Stress Scale (DASS-21; [24]) was used to measure each participant’s fear/anxiety as this variable is positively related to courage (e.g., [34]). The questionnaire included statements about depression, anxiety, and stress, in which participants rated the extent to which each statement applied to them. Rating scale answers ranged from 0 (did not apply to me at all) to 3 (applied to me very much or most of the time). An example statement is “I felt I was close to panic”. Certain items pertain to anxiety, and these items were added and scored to indicate levels of anxiety. The scores from the items were averaged, and Cronbach’s alpha for the anxiety portion of the DASS-21 scale was calculated; it was 0.820.

#### 2.2.4. Sleep

The participants were asked to report the number of hours they slept the previous night. The Pittsburgh Sleep Quality Index (PSQI; [5]) is a questionnaire that evaluates sleep quality and sleep disturbances with open-ended questions and Likert-type responses to statements. The questionnaire uses seven closed-ended ratings on a 4-point Likert-type scale ranging from 1 (Not during the past month) to 4 (Three or more times a week). Furthermore, the participants were given a space to name any disturbances not named by the questionnaire, with a scale to provide the frequency with which the item was experienced. The scores from the items were averaged, and Cronbach’s alpha for the PSQI scale was 0.712. Reliabilities of 0.70 and higher are acceptably consistent according to past research on this issue ([10]; [19]).

#### 2.2.5. Combined Fear/Anxiety and Sleep Measures

The combined fear/anxiety and sleep measures were calculated by multiplying fear/anxiety with each sleep measure. The first measure multiplied fear/anxiety by hours of sleep (quantitative), and the second measure multiplied fear/anxiety by poor sleep quality (qualitative via the modified PSQI).

### 2.3. Procedure

The participants signed up for this study online through the SONA system. As this survey was completed online through Qualtrics, the participants then received an external link that took them to the digital survey. The participants were provided with a written briefing that depicted the risks and benefits of completing the study. All consent procedures concluded with contact information for the university’s counseling center, the study’s primary investigator, the university IRB chair, as well as assurances that all collected data would remain confidential and not tied to their identity. The participants were presented with a series of basic demographic questionnaires, DASS-21, the modified PSQI, and the modified WPCS-23. The WPCS-23 is a two-part scale in which participants are presented with a challenging behavior and asked the degree to which they would willingly agree to engage in the behavior and the fear they would experience when engaging in that behavior. Following completion of each of the questionnaires, the participants were debriefed, and the researchers provided them with credit through SONA.

### 2.4. Design and Analyses

We used a cross-sectional design. We conducted correlational analyses of the individual relations between fear/anxiety, sleep, and the combined relation of anxiety/fear by the sleep measures to the willingness to engage in heroically, daily, and both heroically and daily courageous behaviors, as well as the fear experienced when engaging in those behaviors. We also conducted standard multiple-stepwise regression analyses with all the predictors for each of the six courage measures to determine if anxiety/fear by both sleep measures could predict unique variance in the courage measures above and beyond fear/anxiety and sleep.

## 3. Results

### 3.1. Correlations of Courage Measures and the Four Predictors

Table 1, Table 2, Table 3, Table 4, Table 5, Table 6 and Table 7 present the correlations, and they are listed below. Heroic agreement courage was negatively correlated with hours of sleep, *r*(212) = −0.147, *p* = 0.032, and the Fear/Anxiety × Hours of Sleep interaction, *r*(212) = −0.190, *p* = 0.005. In addition, everyday agreement courage was negatively correlated with fear/anxiety, *r*(210) = −0.236, *p* = 0.001, the Fear/Anxiety × Poor Sleep interaction, *r*(202) = −0.195, *p* = 0.005, and the Fear/Anxiety × Hours of Sleep interaction, *r*(210) = −0.249, *p* = 0.001. Total (heroic and everyday) agreement courage was negatively correlated with hours of sleep, *r*(205) = −0.139, *p* < 0.05, fear/anxiety, *r*(205) = −0.238, *p* = 0.001, the Fear/Anxiety × Poor Sleep interaction, *r*(197) = −0.197, *p* = 0.005, and the Fear/Anxiety × Hours of Sleep interaction, *r*(205) = −0.275, *p* < 0.001.

Fear when engaging in heroically courageous actions was positively correlated with poor sleep quality, *r*(195) = 0.250, *p* < 0.001, fear/anxiety, *r*(204) = 0.209, *p* = 0.003, the Fear/Anxiety × Poor Sleep interaction, *r*(195) = 0.295, *p* < 0.001, and the Fear/Anxiety × Hours of Sleep interaction, *r*(204) = 0.197, *p* = 0.005. Fear when engaging in daily courageous actions was positively correlated with poor sleep quality, *r*(198) = 0.295, *p* < 0.001, fear/anxiety, *r*(206) = 0.401, *p* < 0.001, Fear/Anxiety × Poor Sleep interaction, *r*(198) = 0.414, *p* < 0.001, and the Fear/Anxiety × Hours of Sleep interaction, *r*(206) = 0.371, *p* < 0.001. Fear when engaging in heroically and daily courageous actions was positively correlated with poor sleep quality, *r*(191) = 0.306, *p* < 0.001, fear/anxiety, *r*(199) = 0.345, Fear/Anxiety × Poor Sleep interaction, *r*(191) = 0.397, *p* < 0.001, and the Fear/Anxiety × Hours of Sleep interaction, *r*(199) = 0.318, *p* < 0.001.

### 3.2. Regression Analyses of the Courage Measures Accounted for by the Four Predictors

The results of a stepwise regression Model 1 showed that the Fear/Anxiety x Hours of Sleep interaction, *B* = −0.023 (*SE* = 0.008), *t*(203) = −2.822, *p* = 0.005, negatively predicted 3.8% of the variance in heroic agreement courage. In addition, the results of a stepwise regression Model 2 showed that fear/anxiety, *B* = −0.663 (*SE* = 0.187), *t*(201) = −3.540, *p* < 0.001 negatively predicted 7.6% of the variance, and the Fear/Anxiety × Poor Sleep interaction, *B* = 0.126 (*SE* = 0.062), *t*(201) = 2.038, *p* = 0.043, positively predicted an additional 1.9% of the variance for a total of 9.5% of the variance in everyday agreement courage. Furthermore, the results of a stepwise regression Model 1 with the Fear/Anxiety × Hours of Sleep interaction, *B* = −0.028 (*SE* = 0.007), *t*(197) = −4.186, *p* < 0.001, negatively predicted 8.2% of the variance in total (heroic and daily) agreement courage.

The results of a stepwise regression Model 1 showed that the Fear/Anxiety × Poor Sleep interaction, *B* = 0.126 (*SE* = 0.029), *t*(195) = 4.311, *p* < 0.001, positively predicted 8.7% of the variance in fear when engaging in heroically courageous actions. The results of a stepwise regression Model 1 showed that fear/anxiety, *B* = 0.565 (*SE* = 0.087), *t*(198) = 6.506, *p* < 0.001, positively predicted 17.6% of the variance in fear when engaging in daily courageous actions. The results of a stepwise regression Model 1 with the Fear/Anxiety × Poor Sleep interaction, *B* = 0.157 (*SE* = 0.026), *t*(191) = 5.972, *p* < 0.001, positively predicted 15.7% of the variance in fear when engaging in heroically and daily courageous actions. Figure 1, Figure 2, Figure 3 and Figure 4 visually present the results of four significant interactions predicting courage variables. Whereas Figure 1 displays that sleep is differentially related to courage only at low fear/anxiety levels, Figure 2, Figure 3 and Figure 4 show that sleep differentially predicts courage only at high levels of fear/anxiety. Although the Figure 1 results neither support nor refute the prediction that dysfunctional sleep combines with high fear/anxiety and leads to high courage, the results in Figure 2 and Figure 3 partially support this hypothesis, with the greatest differences in courage due to sleep occurring at high fear/anxiety. Moreover, the results in Figure 4 fully support the interaction hypothesis.

## 4. Discussion

The goal of the current study was to examine whether fear/anxiety would combine with sleep measures (amount in hours and poor sleep quality) to predict unique variance in courage measures. Courage was defined as the willingness to be brave despite fear, and it was measured by willingness to engage in heroic, daily, and both types of courageous acts, as well as fear when engaging in heroic, daily, and both types of courageous acts. Fear/anxiety combined with a sleep measure to predict five of the six courage measures. In addition, these interaction variables were the sole predictor in four of the five analyses and the secondary predictor of courage in one analysis. The figures for four of these interactions provide varying degrees of support for the interaction hypothesis, with the first figure neither supporting nor refuting the hypothesis, the second and third figures partially supporting the hypothesis, and the final figure fully supporting the interaction hypothesis. Therefore, the current study was successful in terms of its main goal.

The findings suggest that sleep alone did not significantly predict courage in regression models, including fear/anxiety and the fear/anxiety by sleep interactions, meaning that sleep did not influence courage on its own. While this result seems to contradict past research showing that sleep deprivation was connected to risky/poor decision-making (e.g., [23]), which is positively related to courage ([4]; [27]), two facts might explain the different results. First, the studies demonstrating a relationship between sleep deprivation and risky decision-making did not examine courage. Second, these studies did not examine sleep along with fear/anxiety and the fear/anxiety by sleep interaction as predictors of risk-taking or courage.

The fact that we did not find a relationship between sleep deprivation and courage in the current study was not surprising because the past literature failed to demonstrate such a relationship. In fact, we had to consult the literature on sleep and risky decision-making (e.g., [23]) and the literature on risky decision-making and courage ([7]; [31]) to suggest that a link might exist between sleep deprivation and courage. Although we did not find a relationship between sleep deprivation and courage in the literature or the current study, we did find that sleep deprivation and fear/anxiety combined to predict courage. Presumably, poor sleep clouded participants’ cognition ([9]; [22]) and helped them ignore high fear levels and provide courageous responses.

As the ratio of women to men favored women by 2 to 1, this difference could have accounted for the results, because women report higher rates of sleep deprivation than men ([42]). Although our ratio pales in comparison to nationwide ratios of nearly 4 to 1, with women representing 78% of undergraduate students and 71% of graduate psychology students ([26] as cited in [15]), it could have been an issue. Therefore, future studies should replicate the current study with 250 men and 250 women and determine if the results differ for men and women, such that those results match the results in the current study better for women than men. The population composition may have influenced sleep deprivation, as college students are known for their chronic sleep deprivation. Future research should replicate and extend the current method using non-college students.

As the measure in the current study evaluated hypothetical behaviors, future research could target actual behaviors that demand courage, such as the BAT spider procedure, the BART balloon procedure, and the Iowa Gambling Task. These studies could determine if the results from the current study on imagined actions extend to actual behaviors. Similar findings in these proposed studies to the results in the current study would provide credence for the speculation that sleep deprivation inhibits cognition to override fear and enhance courageous actions. The participants could also be asked if they think their sleepiness helped reduce their fear and increased their courage. Although participant responses could help confirm the cognitive clouding explanation, they cannot refute it because participants may not be aware of the effect even if it is occurring.

The results of this study can be used to help encourage or reduce courageous behaviors. For example, firefighters and police officers must engage in courageous actions, which could be high in newbies and low in journeymen/journeywomen or vice versa. Novice firefighters or police officers may show low levels of courage, and they should increase them. Alternatively, these individuals may show very high levels of courage, which could heighten their anxiety and impair their decision-making, further increasing their courage. Therefore, the goal would be to teach novices to match the willingness to engage in heroic and everyday courage and fear experienced when engaging in these acts of their supervisors and/or experts in the field.

Based on the results of the current study, sleep deprivation and fear/anxiety could be enhanced to increase courage, or they could be decreased to reduce courage. Many fields may already be ahead of this suggestion, as military personnel, police officers ([23]), firefighters, and emergency room doctors ([4]) are often sleep deprived, and their jobs demand courage. However, achieving higher courage levels than their supervisors or experts in the field could lead to overconfidence ([14]) and poor decision-making ([30]), which could be dangerous for them and the people who count on them to make solid decisions. Therefore, researchers would need to determine the levels of courage experienced by novices and experts in dangerous positions demanding courage using the adapted version of the Woodard-Pury Courage Scale-23 presented in the current study. At that point, courage-increasing or -decreasing programs could be created using relevant fear-provoking situations and techniques that enhance sleep deprivation or sufficient sleep. The individuals could be trained in safe environments rather than learning on the job, which reduces risk for them and the public they serve.

The current study included some limitations. The first limitation of the current study was the sample size. Although we may have found additional effects with a larger sample than the one used in the current study, we found many significant effects that supported the goals and hypotheses in this study. However, future studies could increase the sample size to increase the power and improve the representativeness of the findings. Replication is another way to increase the representativeness and generalization of the results. Similarly, women, as mentioned previously, comprised two-thirds of the sample in the current study; so, future replications could increase the representativeness of the results with large samples containing equal numbers of men and women.

Another limitation was the design of this study, as the current study used a cross-sectional design; these studies can determine correlation, but not causation. Future experiments can manipulate sleep in a sleep lab and fear using movie clips and/or relaxation methods to assess the individual and combined effects of these variables on courage. The randomization of participants to conditions would also control for additional limitations involving habits that increase or reduce stress levels, such as smoking and busy schedules, as well as exercise and meditation, respectively. As shown by [30] ([30]), political leaning strongly influenced determinations of courage in other individuals, which means that it could have affected the results in the current study. Therefore, future research should measure political leaning and other types of beliefs (e.g., religion) and statistically control their influence on courage.

Some relevant factors could have influenced sleep and fear/anxiety, such as caffeine intake, stress, and comorbid mental health conditions (e.g., depression). Caffeine is a widely used stimulant and can disrupt sleep patterns, especially when consumed before sleeping ([11]). In addition, caffeine increases anxiety ([18]). Therefore, caffeine consumption could have influenced these predictors and their individual and combined relations to courage ([43]). In addition, psychological distress comorbid with anxiety, such as depression, could influence sleep ([1]), complicating the relationship between sleep deprivation and courage. Future studies should control these variables when replicating the current study.

In summary, the goals of the current study were achieved, and the hypotheses were supported. Specifically, poor sleep (in hours and overall sleep quality) combined with fear/anxiety and predicted unique variance for both everyday courage and heroic courage. Although the current study successfully found that fear/anxiety combined with sleep measures to predict various measures of courage, future research should determine the mechanisms behind these findings. Further study on this topic may help researchers understand the optimal levels of courage in different settings, as well as methods to achieve those levels of courage. To potentially apply these findings to real-world settings, future research would first need to extend the procedures in the current study to novices and experts in particular fields (e.g., safety and medicine) and manipulate sleep and fear/anxiety to make novices more like experts. Future research should also control for personal beliefs, such as political leaning. Although the idea of being a fearless superhero seems enticing at first blush, the current study showed that we need fear to overcome, but we also may need a little brain fog from sleep deprivation to jump into threatening situations. Therefore, the conclusion of this work for now is that, as Captain Kirk told Bones in *Star Trek: The Final Frontier*, fear is needed as it helps us survive, defines us, reveals our limitations and abilities, and it also combines with poor sleep to stay our courage.

## Figures and Tables

**Figure 1 behavsci-15-00634-f001:**
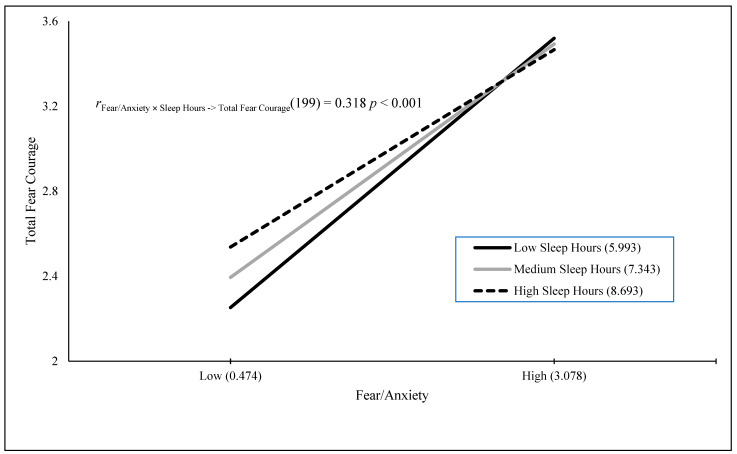
Fear/anxiety predicting total fear courage across sleep hours.

**Figure 2 behavsci-15-00634-f002:**
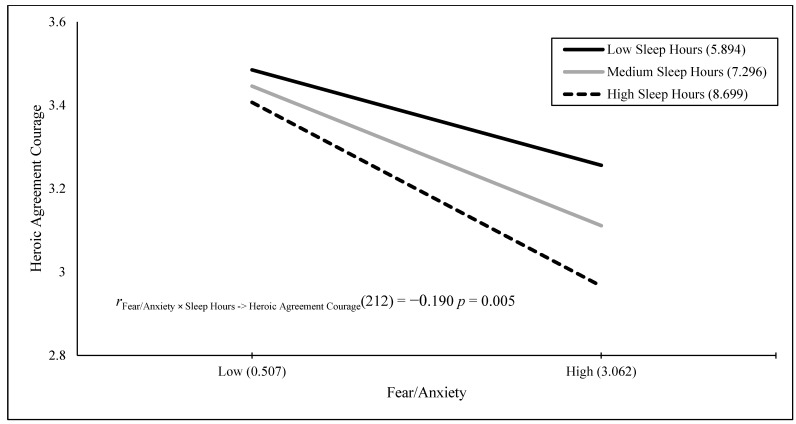
Fear/anxiety predicting heroic agreement courage across sleep hours.

**Figure 3 behavsci-15-00634-f003:**
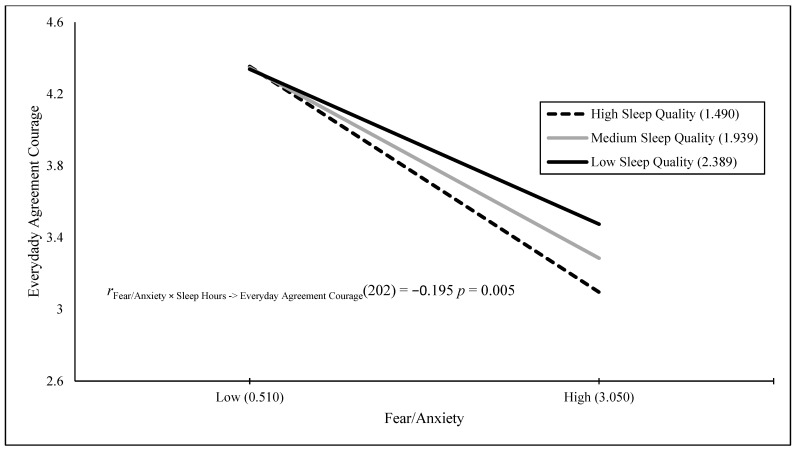
Fear/anxiety predicting everyday agreement courage across sleep quality.

**Figure 4 behavsci-15-00634-f004:**
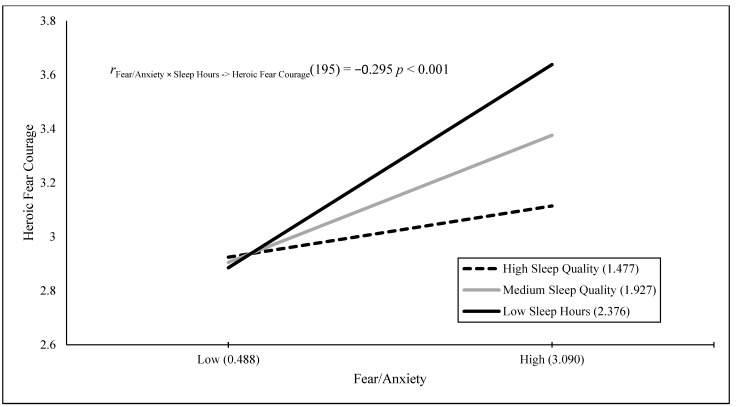
Fear/anxiety predicting heroic fear courage across sleep quality.

**Table 1 behavsci-15-00634-t001:** Pearson’s correlation matrix of the Hours Sleep, Poor Sleep, Fear/Anxiety, Poor Sleep x Fear, Hours Sleep x Fear predictors and Heroic Agreement Courage, Everyday Agreement Courage, Total Agreement Courage, Fear of Heroic Acts of Courage, Fear of Everyday Acts of Courage, and Fear of All Acts of Courage criteria.

	1	2	3	4	5	6	7	8	9	10	11
1. Hrs Sleep (HS)	—										
2. Poor Sleep (PS)	**−0.172 ** (223)**	—									
3. Fear/Anx	0.001 (232)	**0.541 ** (223)**	—								
4. PS × Fear	−0.043 (223)	**0.808 ** (223)**	**0.914 ** (223)**	—							
5. HS × Fear	**0.493 ** (232)**	**0.373 ** (223)**	**0.856 ** (232)**	**0.751 ** (223)**	—						
6. Hero Agr Cour	**−0.147 * (214)**	−0.001 (205)	−0.128 (214)	−0.101 (205)	**−0.190 ** (214)**	—					
7. Evry Agr Cour	−0.100 (212)	−0.046 (204)	**−0.236 ** (212)**	**−0.195 ** (204)**	**−0.249 ** (212)**	**0.423 ** (207)**	—				
8. Total Arg All	**−0.139 * (207)**	−0.046 (199)	**−0.238 ** (207)**	**−0.197 ** (199)**	**−0.275 ** (207)**	**0.874 ** (207)**	**0.793 ** (199)**	—			
9. Fear Hero Cour	0.039 (206)	**0.250 ** (197)**	**0.209 **(206)**	**0.295 ** (197)**	**0.197 ** (206)**	**−0.145 * (206)**	−0.134 (200)	**−0.147 * (200)**	—		
10. Fear Evry Cour	0.060 (208)	**0.295 ** (200)**	**0.401 ** (208)**	**0.414 ** (200)**	**0.371 ** (208)**	−0.117 (205)	**−0.371 ** (207)**	**−0.264 ** (204)**	**0.683 ** (201)**	—	
11. Fear All Cour	0.053 (201)	**0.306 ** (193)**	**0.345 ** (201)**	**0.397 ** (193)**	**0.318 ** (201)**	−0.133 (201)	**−0.258 ** (200)**	**−0.214 ** (200)**	**0.935 ** (201)**	**0.897 ** (201)**	—

*p*(*N*); *** *p* ≤ 0.05**, **** *p* ≤ 0.01**; 1. Hours of Sleep, 2. Poor Sleep, 3. Fear/Anxiety, 4. Poor Sleep × Fear/Anxiety, 5. Hours of Sleep × Fear/Anxiety, 6. Heroic Agreement Courage, 7. Everyday Agreement Courage, 8. Total Agreement Courage, 9. Fear of Heroic Acts of Courage, 10. Fear of Everyday Acts of Courage, 11. Fear of All Acts Courage.

**Table 2 behavsci-15-00634-t002:** Pearson’s correlation matrix of the Hours of Sleep, Poor Sleep, Fear/Anxiety, Poor Sleep × Fear, Hours Sleep × Fear predictors and Heroic Agreement Courage criteria.

	1	2	3	4	5
6. Hero Agr Cour	**−0.147 * (214)**	−0.001 (205)	−0.128 (214)	−0.101 (205)	**−0.190 ** (214)**

*p*(*N*), *** *p* < 0.05**, **** *p* < 0.01**; 1. Hours of Sleep, 2. Poor Sleep, 3. Fear/Anxiety, 4. Poor Sleep × Fear/Anxiety, 5. Hours of Sleep × Fear/Anxiety, 6. Heroic Agreement Courage.

**Table 3 behavsci-15-00634-t003:** Pearson’s correlation matrix of the Hours of Sleep, Poor Sleep, Fear/Anxiety, Poor Sleep × Fear, Hours Sleep × Fear predictors and Everyday Agreement Courage criteria.

	1	2	3	4	5
7. Evry Agr Cour	−0.100 (212)	−0.046 (204)	**−0.236 ** (212)**	**−0.195 ** (204)**	**−0.249 ** (212)**

*p*(*N*), **** *p* < 0.01**; 1. Hours of Sleep, 2. Poor Sleep, 3. Fear/Anxiety, 4. Poor Sleep × Fear/Anxiety, 5. Hours of Sleep × Fear/Anxiety, 7. Everyday Agreement Courage.

**Table 4 behavsci-15-00634-t004:** Pearson’s correlation matrix of the Hours Sleep, Poor Sleep, Fear/Anxiety, Poor Sleep × Fear, Hours Sleep × Fear predictors and Total Agreement Courage criteria.

	1	2	3	4	5
8. Total Arg All	**−0.139 * (207)**	**−0.046 (199)**	**−0.238 ** (207)**	**−0.197 ** (199)**	**−0.275 ** (207)**

*p*(*N*), *** *p* < 0.05**, **** *p* < 0.01;** 1. Hours of Sleep, 2. Poor Sleep, 3. Fear/Anxiety, 4. Poor Sleep × Fear/Anxiety, 5. Hours of Sleep × Fear/Anxiety, 8. Total Agreement Courage.

**Table 5 behavsci-15-00634-t005:** Pearson’s correlation matrix of the Hours of Sleep, Poor Sleep, Fear/Anxiety, Poor Sleep × Fear, Hours Sleep × Fear predictors and Fear of Heroic Acts of Courage criteria.

	1	2	3	4	5
9. Fear Hero Cour	0.039 (206)	**0.250 ** (197)**	**0.209 ** (204)**	**0.295 ** (197)**	**0.197 ** (206)**

*p*(*N*), **** *p* < 0.01**; 1. Hours of Sleep, 2. Poor Sleep, 3. Fear/Anxiety, 4. Poor Sleep × Fear/Anxiety, 5. Hours of Sleep × Fear/Anxiety, 9. Fear of Heroic Acts of Courage.

**Table 6 behavsci-15-00634-t006:** Pearson’s correlation matrix of the Hours of Sleep, Poor Sleep, Fear/Anxiety, Poor Sleep × Fear, Hours Sleep × Fear predictors and Fear of Everyday Acts of Courage criteria.

	1	2	3	4	5
10. Fear Evry Cour	0.060 (208)	**0.295 ** (200)**	**0.401 ** (208)**	**0.414 ** (200)**	**0.371 ** (208)**

*p*(*N*), **** *p* < 0.01**; 1. Hours of Sleep, 2. Poor Sleep, 3. Fear/Anxiety, 4. Poor Sleep × Fear/Anxiety, 5. Hours of Sleep × Fear/Anxiety, 10. Fear of Everyday Acts of Courage.

**Table 7 behavsci-15-00634-t007:** Pearson’s correlation matrix of the Hours of Sleep, Poor Sleep, Fear/Anxiety, Poor Sleep × Fear, Hours Sleep × Fear predictors and Fear of All Acts of Courage criteria.

	1	2	3	4	5
11. Fear All Cour	0.053 (201)	**0.306 ** (193)**	**0.345 ** (201)**	**0.397 ** (193)**	**0.318 ** (201)**

*p*(*N*), **** *p* < 0.01**; 1. Hours of Sleep, 2. Poor Sleep, 3. Fear/Anxiety, 4. Poor Sleep × Fear/Anxiety, 5. Hours of Sleep × Fear/Anxiety, 11. Fear of All Acts Courage.

## Data Availability

The raw data supporting the conclusions of this article will be made available by the authors upon reasonable request.

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
