# Peer review of "Fear/Anxiety and Sleep Deprivation Combine to Predict Courage"

_behavsci, 2025, doi:10.3390/bs15050634_

Round 1

Reviewer 1 Report

Comments and Suggestions for Authors

My major concerns about this study, in order from most serious to least, are:

The description of the WPCS-23 is flawed and is not an accurate representation of that scale. For example, the individual item "I would seek out and ask a grocery store employee for help when shopping" does not appear in the scale and appears to have been developed solely for this study. While it is fine to use adapted versions of scales in research, it is crucial that readers are informed that 1) adaptations were made and 2) what those adaptations are. 

I compared the list of items presented in the original WPCS - 23 article with the item numbers described as belonging to each of the categories by the authors on p. 6. What I got were items that did not track what was presented in the article:

Heroic items: 

1) I would accept an important project at my place of employment even though it would bring intense public criticism and publicity.
2) If it looked like someone would get badly hurt, I would intervene directly in a dangerous domestic dispute.
3) I could approach someone whose family members had just been killed, knowing they were feeling overwhelming grief.
5) If called upon during times of national emergency, I would give my life for my country
6) I am able to participate in intense conflict in a work environment for the right cause
7) I would talk to my supervisor about a raise if I really needed one.
8) I would go to the dentist and have painful surgery if it meant saving a tooth.
9) I would risk my life if it meant lasting world peace.
11) I would refuse the order of a commanding officer if it meant hurting someone needlessly.
12) I could do without the absolute necessities of life if there were others in greater need.

Everyday items:

4) I would risk rejection by important others for a chance at achieving my life goals.
10) Intense social pressure would not stop me from doing the right thing.
13) I would confront a parent abusing his or her child in public.
14) I would walk across a dangerously high bridge to continue on an important journey.
15) I would endure physical pain for my religious or moral beliefs. 
17) I would open myself to professional criticism by publishing my work.
18) I could move to a foreign country to have the perfect job.
19) I could keep my wits about me if I were lost in the woods at night.

Somewhere along the line, something that was not the original scale appears to have been used instead. However, given this it is difficult to know what to make of the rest of the article.

The term "personal courage" is the term that has been used in past literature for what the authors call "individual courage" and "everyday courage" and it should be used here to prevent confusion.

That term also brings another possible issue to light: the types of risks in the scale used to measure fear and courage differ - some are risks of physical pain and danger and some are risks of social disapproval or discomfort. Is it possible that sleep deprivation differentially affects willingness to take different types of risks? For example, I know that I am more clumsy when sleep deprived but also I may simply care less what others think of me. Without the items being divided in that way, however, it is impossible to know how this could have affected results. 

I found the results, presented only in tabular form and in the text, to be difficult to understand. A graph or two illustrating the interaction would be useful.

Author Response

We appreciate all the comments from the reviewers because we know that they will enhance the readability and quality of the paper. Reviewer 1 said “My major concerns about this study, in order from most serious to least, are: The description of the WPCS-23 is flawed and is not an accurate representation of that scale. For example, the individual item "I would seek out and ask a grocery store employee for help when shopping" does not appear in the scale and appears to have been developed solely for this study. While it is fine to use adapted versions of scales in research, it is crucial that readers are informed that 1) adaptations were made and 2) what those adaptations are.”  Reviewer 1 also said “I compared the list of items presented in the original WPCS - 23 article with the item numbers described as belonging to each of the categories by the authors on p. 6. What I got were items that did not track what was presented in the article:” and they listed the items.  Reviewer 1 also said “Somewhere along the line, something that was not the original scale appears to have been used instead. However, given this it is difficult to know what to make of the rest of the article.” We addressed these comments by providing the WPCS-23 items that we used in Appendix A along with any new wording and new items.  Appendix B presents the new adapted questionnaire that we used in the current study in its entirety.  

Reviewer 1 said “The term "personal courage" is the term that has been used in past literature for what the authors call "individual courage" and "everyday courage" and it should be used here to prevent confusion.”  As Reviewer 1 stated, we made many changes to this scale, and we believe that the term “everyday courage” captures the essence of these statements/items better than “personal courage” because many items that comprise the “heroic courage” portion of the scale are also personal acts of courage. The issue that separates the two groups is the frequency and commonality of the items. The everyday statements/items pertain to acts that could/would happen much more frequently (i.e., everyday) than the heroic statements/items (i.e., rarely), as the heroic statements demand much more courage and elicit more anxiety/fear.  We would certainly be open to calling the everyday courage items “everyday personal courage” and the heroic items “rare heroic courage” if Reviewer 1 prefers this terminology.

Reviewer 1 also said “That term also brings another possible issue to light: the types of risks in the scale used to measure fear and courage differ - some are risks of physical pain and danger and some are risks of social disapproval or discomfort”. If Reviewer 1 wants us to suggest that future research should examine these issues using our scale in the future, we are open to that idea.

Reviewer 1 said “Is it possible that sleep deprivation differentially affects willingness to take different types of risks? For example, I know that I am more clumsy when sleep deprived but also I may simply care less what others think of me. Without the items being divided in that way, however, it is impossible to know how this could have affected results.”  I am not sure what way that Reviewer 1 is suggesting to divide the items.  Future research could certainly examine more participants and try to use our scale or a new adaption to address this issue as the possibility that Reviewer 1’s example and question are interesting and a potential direction for future research.

Reviewer 1 said “I found the results, presented only in tabular form and in the text, to be difficult to understand. A graph or two illustrating the interaction would be useful.”  We addressed this issue by presenting the entire correlation table (Table 1) and then presenting 6 additional tables (Tables 2-7) to show the predictors for each measure of courage.  We hope these changes corrected the problem.  If Reviewer 1 still wants us to create separate plots of the interaction effects, I will learn how to make those figures.  I have a colleague who can show me how to use midpoint plotting to demonstrate the effect as if it were dichotomous variables.  Please let us know if the current presentation is insufficient and I need to make those figures.  We would like to thank Reviewer 1 for all of their helpful comments and any future ones they will make.

Reviewer 2 Report

Comments and Suggestions for Authors

This manuscript explores the interaction between fear/anxiety and sleep deprivation on courage, addressing a gap in existing research. The hypotheses are well-founded, and the methodology is rigorous. However, several issues require clarification and expansion to strengthen the study’s validity and overall impact.

Major comments:

  1. The gender imbalance (66% women) may limit the generalizability of findings, particularly given prior research suggesting sex differences in sleep-deprivation effects. The authors should discuss how this imbalance might influence results and explain why such a sex imbalance occurred in the sample selection.

  1. The modified WPCS-23 removed two items and underwent factor analysis, but insufficient details are provided (e.g., factor loadings, model fit indices). Please clarify how the modified scale was validated and whether it retains the original’s psychometric properties.

The Pittsburgh Sleep Quality Index (PSQI) had a Cronbach’s alpha of .712, which is below conventional thresholds for reliability. Please give potential implications of this for interpreting sleep quality results.

  1. The finding that sleep deprivation alone has no effect on courage contradicts previous research that associates sleep loss with increased risk-taking behavior. Please provide the rationale for why sleep’s effect is only evident when interacting with fear/anxiety.

  1. Variables such as caffeine intake, stress levels, or comorbid mental health conditions (e.g., depression) were not controlled. Discuss how these factors might confound the sleep-courage relationship.

Minor comments:

  1. Table 1 is difficult to interpret. Consider reformatting for readability, like using bold text for significance markers.

  1. Ensure all cited works are peer-reviewed and available to public. Some references are incomplete in the text; please verify consistency with the reference list.

Comments on the Quality of English Language

The English writing is great.

Author Response

We appreciate the comments made by Reviewer 2.  Reviewer 2 said “This manuscript explores the interaction between fear/anxiety and sleep deprivation on courage, addressing a gap in existing research. The hypotheses are well-founded, and the methodology is rigorous”. We would like to thank Reviewer 2 for this compliment.  Reviewer 2 also said “However, several issues require clarification and expansion to strengthen the study’s validity and overall impact.”  Reviewer 2 said that their major comments pertain to gender imbalance and the factor analysis. 

Specifically, Reviewer 2 said “The gender imbalance (66% women) may limit the generalizability of findings, particularly given prior research suggesting sex differences in sleep-deprivation effects. The authors should discuss how this imbalance might influence results and explain why such a sex imbalance occurred in the sample selection.”  We thank Reviewer 2 for this suggestion and we followed it.

Reviewer 2 also said that “The modified WPCS-23 removed two items and underwent factor analysis, but insufficient details are provided (e.g., factor loadings, model fit indices). Please clarify how the modified scale was validated and whether it retains the original’s psychometric properties.” These details are not provided in the preliminary analyses.

Reviewer 2 also made points about the reliability of the PSQI, the lack of a relation between sleep and courage in the study, and influential variables.  Specifically, Reviewer 2 said “The Pittsburgh Sleep Quality Index (PSQI) had a Cronbach’s alpha of .712, which is below conventional thresholds for reliability. Please give potential implications of this for interpreting sleep quality results.”  We cited research stating that reliabilities of .7 are sufficient.

Reviewer 2 also said “The finding that sleep deprivation alone has no effect on courage contradicts previous research that associates sleep loss with increased risk-taking behavior. Please provide the rationale for why sleep’s effect is only evident when interacting with fear/anxiety.” We made this point in the Discussion and addressed it by the fact that we could not find literature linking sleep deprivation and courage. Instead, we found literature linking sleep deprivation and risk-taking and literature linking risk-taking and courage, so that we could link sleep deprivation with courage indirectly. Basically, the literature did not cite the relation because it did not exist in past literature and it was not found in our study either.

In addition, Reviewer 2 said “Variables such as caffeine intake, stress levels, or comorbid mental health conditions (e.g., depression) were not controlled. Discuss how these factors might confound the sleep-courage relationship.”  In the Discussion, we addressed the potential effects of caffeine and comorbid psychological distress, such as depression.

Reviewer 2 made minor comments about the legibility of Table 1 and cited works.  Specifically, Reviewer 2 said “Table 1 is difficult to interpret. Consider reformatting for readability, like using bold text for significance markers.”  As stated earlier, we provided Table 1 and 6 additional tables showing the relation of each dependent variable to the predictors.

Reviewer 2 also said “Ensure all cited works are peer-reviewed and available to public. Some references are incomplete in the text; please verify consistency with the reference list.”  We made corrections to the references.

Round 2

Reviewer 1 Report

Comments and Suggestions for Authors

Please stick with consistent labels of your constructs throughout. This should be true for past literature (Pury et al. 2007 does not use the term "individual courage" - the term is "personal courage") as well as your own proposed distinction between heroic courage and everyday courage, which should be clearly defined in the introduction and distinguished from the terms already established in the literature (general courage and personal courage). It's fine if there is a distinction to be made, but for the sake of future researchers looking for research on a construct, it is important to be clear what those constructs are and how they differ from past, similar constructs. If there is no difference, the existing labels should be used. 

The abstract needs additional clarification on "a version of the Woodard Pury Courage Scale-23", which sounds like you've used a variant that can be found in that paper. Instead, you've created a new scale that includes about 50% revised or entirely new items. Transparency on this issue in the abstract to me would look like "We developed a modified version of the WPCS 23 that included items with a clear distinction between heroic and everyday courageous acts." As noted above though, I think those items are better characterized as general and personal courage unless you have a compelling reason for new termenology. 

Finally, I do think you need to include additional statistical tests and supporting graphs to illustrate what the interactions that you found mean. 

Author Response

April 21, 2025

Editor

Behavioral Sciences

We are resubmitting a manuscript entitled, “Fear/Anxiety and Sleep Deprivation Combine to Predict Courage” for publication in Behavioral Sciences. This paper evaluates the combined relation of fear/anxiety and sleep deprivation to six courage measures, which has not been demonstrated in the literature previously. The results showed that five of those courage measures are predicted by the interaction created by the product of fear/anxiety and sleep deprivation.

The editor said “Please stick with consistent labels of your constructs throughout. This should be true for past literature (Pury et al. 2007 does not use the term "individual courage" - the term is "personal courage") as well as your own proposed distinction between heroic courage and everyday courage, which should be clearly defined in the introduction and distinguished from the terms already established in the literature (general courage and personal courage). It's fine if there is a distinction to be made, but for the sake of future researchers looking for research on a construct, it is important to be clear what those constructs are and how they differ from past, similar constructs. If there is no difference, the existing labels should be used. The abstract needs additional clarification on "a version of the Woodard Pury Courage Scale-23", which sounds like you've used a variant that can be found in that paper. Instead, you've created a new scale that includes about 50% revised or entirely new items. Transparency on this issue in the abstract to me would look like "We developed a modified version of the WPCS 23 that included items with a clear distinction between heroic and everyday courageous acts." As noted above though, I think those items are better characterized as general and personal courage unless you have a compelling reason for new termenology. Finally, I do think you need to include additional statistical tests and supporting graphs to illustrate what the interactions that you found mean.” We address these comments below.

My coauthors and I decided that our paper examines everyday courage and heroic courage, not personal and general courage. Therefore, we referenced the WPCS-23 by Pury et al. (2007), because we changed wording in the scale (e.g., saying would for items), added to it, and removed items from it. We said that we adapted the WPCS-23, but we changed more of the questionnaire than we did not change, so the claim may be an understatement. However, the everyday courage and heroic courage ideas came from their questionnaire, so these authors deserve credit for it. We made the distinction clear in the introduction, and we consistently used the terms everyday courage and heroic courage throughout the remainder of the paper. We referred to the adapted WPCS-23 in the abstract. We added graphs of four interactions to the paper. We also found mistakes in the correlation tables, so they have been edited with the correct information. We believe that these changes enhanced the quality of the paper, and it is ready for publication.

All the authors have approved the manuscript and agree with its submission to Behavioral Sciences. I will serve as the corresponding author for this manuscript. I will keep my coauthors informed of the progress through the editorial review process, the results of the reviews, and any revisions to be made. I can be contacted at Christopher Newport University, Department of Psychology, 1 Avenue of the Arts, Newport News, VA 23606; jgibbons@cnu.edu. My telephone number is (757) 594-7256 (FAX: 757-594-7342). Brenna McManus can be contacted at brenna.mcmanus.22@cnu.edu, Ella White can be contacted at ella.white.22@cnu.edu, and Akihaya Gibbons can be contacted at akig@vt.edu. We will be happy to provide biographies upon publication acceptance and your journal should already have a biography for me. We look forward to another thorough review that will lead to the publication of this work.

Sincerely,

Jeffrey A. Gibbons

Associate Professor of Psychology

Christopher Newport University

Round 3

Reviewer 1 Report

Comments and Suggestions for Authors

This is better, but the description of the modifications of the WPCS-23 still read as problematic to me. 

1) Individual courage is your own term, or at least not mine. I've used the term "personal courage" throughout although I think we mean the same thing. Please either use the established term or make an argument for your preferred term.

2) You are setting up a distinction between heroic courage and everyday courage, which is a great area to study. The original WPCS-23 appears to contain a mix of heroic and everyday courage items with the bulk of the items representing heroic courage. You've adapted the scale by adding to the pool of everyday items. You've also modified the wording of several items to improve clarity.

Author Response

April 26, 2025

Editor

Behavioral Sciences

We are resubmitting a manuscript entitled, “Fear/Anxiety and Sleep Deprivation Combine to Predict Courage” for publication in Behavioral Sciences. This paper evaluates the combined relation of fear/anxiety and sleep deprivation to six courage measures, which has not been demonstrated in the literature previously. The results showed that five of those courage measures are predicted by the interaction created by the product of fear/anxiety and sleep deprivation.

The editor/reviewer said, “This is better, but the description of the modifications of the WPCS-23 still read as problematic to me. 1) Individual courage is your own term, or at least not mine. I've used the term "personal courage" throughout although I think we mean the same thing. Please either use the established term or make an argument for your preferred term.”

We are glad that the editor/reviewer liked the last version of the paper better. We used the label “individual courage” once in the previous version of the paper, and we replaced it with “personal courage” in the current version of the paper. We apologize for not using the proper label, which is personal courage. We thought that the editor/reviewer was making a point about general and personal courage being better descriptors/labels than heroic and personal courage. We apologize for our confusion.   

The editor/reviewer said, “2) You are setting up a distinction between heroic courage and everyday courage, which is a great area to study. The original WPCS-23 appears to contain a mix of heroic and everyday courage items with the bulk of the items representing heroic courage. You've adapted the scale by adding to the pool of everyday items. You've also modified the wording of several items to improve clarity.”

We believe that the editor/reviewer perfectly summarized our article’s contribution to the literature. We appreciate their support. We believe that the changes made based on the editor’s/author’s suggestions enhanced the quality of the paper, and it is ready for publication.

All the authors have approved the manuscript and agree with its submission to Behavioral Sciences. I will serve as the corresponding author for this manuscript. I will keep my coauthors informed of the progress through the editorial review process, the results of the reviews, and any revisions to be made. I can be contacted at Christopher Newport University, Department of Psychology, 1 Avenue of the Arts, Newport News, VA 23606; jgibbons@cnu.edu. My telephone number is (757) 594-7256 (FAX: 757-594-7342). Brenna McManus can be contacted at brenna.mcmanus.22@cnu.edu, Ella White can be contacted at ella.white.22@cnu.edu, and Akihaya Gibbons can be contacted at akig@vt.edu. We will be happy to provide biographies upon publication acceptance and your journal should already have a biography for me. We look forward to another thorough review that will lead to the publication of this work.

Sincerely,

Jeffrey A. Gibbons

Professor of Psychology

Christopher Newport University